# Learning Expert-Interpretable Programs for Myocardial Infarction Localization

**Joshua Flashner**
Caltech

**Jennifer Sun**
Caltech

**David Ouyang**
Cedars-Sinai

**Yisong Yue**
Caltech

## Abstract

We study how to learn accurate and interpretable models for assisted clinical diagnostics. We focus on myocardial infarction (heart attack) localization from electrocardiogram (ECG) signals, which is known to have a complex mapping that is challenging even for expert cardiologists to understand. Our approach leverages recent advances in learning neurosymbolic models, and yields inherently expert-interpretable programs as compositions of ECG features and learned temporal filters. We evaluate our method on a set of 21,844 ECG recordings, to localize myocardial infarction at different levels of granularity. Results demonstrate that our model performs comparably to conventional black-box baselines, but with a much simpler and more interpretable structure.

## 1 Introduction

Electrocardiograms (ECGs) measure the electrical activity of the heart, providing cardiologists valuable information to diagnose and treat a range of heart conditions including arrhythmias and myocardial infarction. This waveform data is a complex representation of the heart's electrical activity over different locations, and many factors affect the appearance of the recordings. Thus, there is strong motivation to derive reliable and interpretable models for ECG-based clinical diagnostics.

Like other health domains, interest in utilizing data-driven models like neural networks for ECG classification is growing [21, 22, 8]. However, these "black-box" models are high dimensional and challenging to interpret. One common approach involves using post-hoc explanations of a neural network's predictions, but such methods have limited utility in crucial areas like clinical diagnostics [24].

In this paper, we aim to create data-driven models for ECG analysis that are both accurate and interpretable. We build upon recent advancements in neurosymbolic modeling, which combines neural networks with symbolic components, integrating the flexibility of the former and the semantic interpretability of the latter [4]. We employ neurosymbolic program synthesis [25] to derive interpretable programs from ECG data within a domain-specific language (DSL). Our work explores how these models can encode useful features from cardiologists in the DSL and learn expert-interpretable programs for ECG classification.

We ground our research in the task of Myocardial Infarction (MI) localization from a 12-lead ECG. This fine-grained localization is often challenging for experts but is vital for determining the optimal course of treatment. For example, locating the MI in the left anterior descending artery may lead doctors to recommend percutaneous coronary intervention.

37th Conference on Neural Information Processing Systems (NeurIPS 2023).

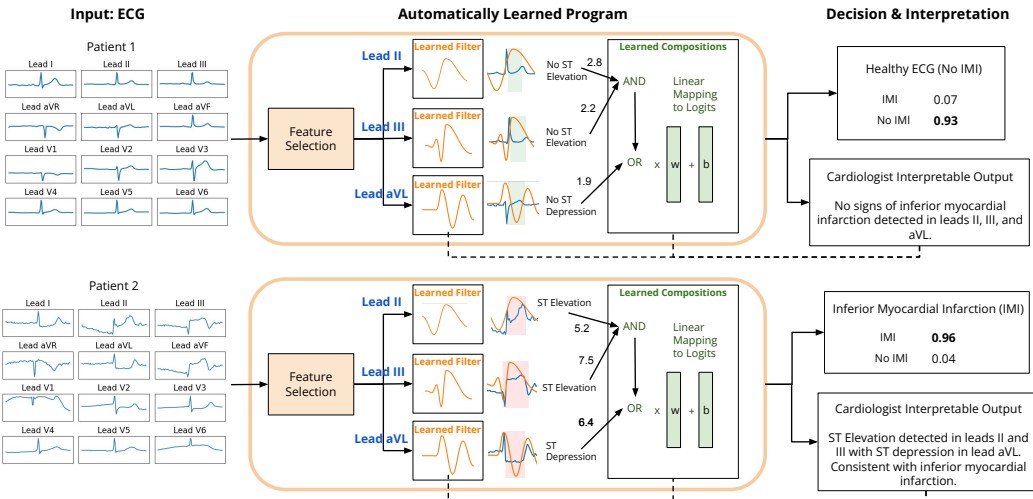

Figure 1: We propose a framework for synthesizing expert-interpretable programs that localize myocardial infarction from 12-lead ECGs. The top row shows an example of a healthy ECG as the input to our automatically learned program while the second row shows an ECG with Inferior Myocardial Infarction (IMI). First, the program selects relevant leads given segmented heartbeats from each lead (*left*). Then, the program applies learned temporal filters to the selected leads, where red highlighting denotes high activation, while green denotes low activation, and numbers correspond to the quantitative measurement of activation (*middle*). The outputs of the temporal filters are passed to a set of learned compositional layers, then linearly mapped to output logits (*right*). The right column shows the final diagnosis and a possible interpretation.

To summarize, this paper develops a framework for learning expert-interpretable models for MI localization from ECG data. By designing a DSL that encodes rich domain knowledge and utilizing recent program synthesis techniques [25], we show that our programs achieve performance comparable to conventional neural network approaches while providing more interpretable outputs that correlate with cardiologist-recognizable MI features.

## 2   Related Work

**ECG Data Analysis.** Recent work studying ECG data has focused on the use of deep learning methods, such as convolutional neural networks (CNNs), for classifying a wide range of heart conditions from ECG data [22, 1, 26, 14, 9]. For example, [21] uses CNNs to classify heart arrhythmias from a single-lead ECG, [18] uses 1D Conv Nets to predict post-procedural mortality from 12-lead ECGs, and [31] reviews a range of methods for MI detection and localization in ECG data, finding that CNNs are the most popular models. Other machine learning techniques, such as contrastive learning [8] and transfer learning [30], have also been explored to improve the accuracy and data efficiency of training. To our knowledge, program synthesis and neurosymbolic modeling have not been well-explored for studying ECG data.

**Interpretable Machine Learning.** There are two main approaches to interpretable machine learning: by explaining black-box models after they are built [17, 23, 12], or by building models that are inherently-interpretable [16, 25, 10, 28, 3]. Our work falls in the second group.

In general, open questions exist on when machine learning models are considered interpretable [24, 11, 15, 13]. In our work, we focus on designing models that are interpretable by domain experts such as cardiologists. Key to our approach is designing a domain specific language (DSL) that leverages features designed by cardiologists, and learning programs or models from that language automatically from data.

**Neurosymbolic Modeling.** Neurosymbolic modeling is a growing research direction [4], and has shown promising results in a range of scientific domains [5, 27]. Existing approaches have

studied training these models with supervised learning [25, 6], with policy learning and reward functions [29, 2], or with generative modeling [7, 32]. We instantiate our work using the approach from [25] for learning differentiable programs, and use a DSL that we designed for ECG data analysis. Our model is trained in a supervised way for MI localization. Our framework will benefit from future improvements in differentiable program synthesis.

# 3  Dataset

## 3.1  Overview

We use a dataset of 12-lead electrocardiograms (ECG) originating from Cedars-Sinai Medical Center (visualized in Figure 2). Our experiments were performed on 21,844 ECGs, where the ECGs are collected within one week of an angiogram. The data was subsequently partitioned into training (80%, 17,475 ECGs), validation (10%, 2,184 ECGs), and testing (10%, 2,185 ECGs) sets. Demographics information of our dataset is in Table 1.

| Lead(s) | Correlated Heart Region |
|---|---|
| I, aVL | Lateral wall of the left ventricle |
| II, III, aVF | Inferior wall of the left ventricle |
| V1, V2 | Interventricular septum and right ventricle |
| V3, V4 | Anterior wall of the left ventricle |
| V5, V6 | Lateral wall of the left ventricle (lower section near apex) |

Figure 2: ECG Leads and Their Correlated Heart Regions. Image courtesy of [20].

## 3.2  Classification Tasks and Label Distributions

We consider three classification tasks, each with a different level of granularity: a binary classification task for myocardial infarction vs no myocardial infarction, a coarse 3 location task localizing to the three main coronary arteries, and a fine grain classification with additional detail with regard to proximal vs. distal disease across multiple branches and 17 locations. The coarse labels are derived from the fine grain labels by grouping them into the three main coronary arteries. The locations used for the coarse task and label distributions are: right coronary artery(RCA) 19%, left anterior descending artery(LAD) 25%, and left circumflex artery(LCX) 14%, normal (no MI) 42%. Locations and labels distributions for the fine grain task can be found in Appendix D.

Table 1: Demographics of our dataset.

| Age, years (SD) | 65.1 (15.9) |
|---|---|
| Female | 45.2% |
| Heart Failure | 18.3% |
| Diabetes | 16.6% |
| Hypertension | 34.8% |
| Coronary Artery Disease | 21.7% |
| Stroke | 5.7% |
| Renal Disease | 9.5% |

# 4  Synthesizing Expert-Interpretable Programs

The key idea is to search for programs that look like the example on the right side of Figure 3. This type of program is essentially a flow-chart, with each individual component being expert-interpretable. In order to find, or synthesize, such a program automatically from data, we require two ingredients:

- A domain specific language (DSL) that includes expert-interpretable functions (i.e., components or modules). Such a DSL is typically designed in collaboration with domain experts, and enables instantiating sparse programs like the flow-chart in Figure 3. Note that individual components can have continuous parameters.

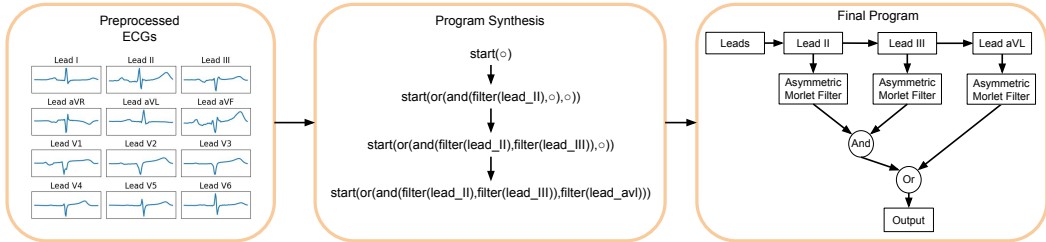

Figure 3: Overview of our framework for learning expert-interpretable programs. First ECGs are preprocessed as shown in Figure 7 of the appendix. These ECGs are used as input for program synthesis (*left*). We instantiate our method using an existing program synthesis approach called NEAR [25], which uses a neural heuristic to search the program space (*middle*). We visualize the final learned program structure (*right*).

- A synthesis algorithm that searches for the best program architecture (flow-chart structure) and continuous parameter settings from data. This step is analogous to neural architecture search, but over a space of programs defined by the DSL.

We discuss in Sections 4.1 & 4.2 our synthesis approach and DSL design, respectively, In Section 4.3, we provide more intuition on how experts can interpret programs from our DSL.

## 4.1 Differentiable Program Synthesis

Following the setup in [25], a program is described in a domain-specific language (DSL) as $(\alpha, \theta)$, where $\alpha$ is the discrete program architecture and $\theta$ corresponds to the real-valued parameters. Our learning objective is:

$$(\alpha^*, \theta^*) = \arg\min_{(\alpha, \theta)} (s(\alpha) + \zeta(\alpha, \theta)).$$

$\zeta(\alpha, \theta) = \mathbb{E}_{(x,y) \sim D}[1(\alpha_\theta(x) \neq y)]$ is the standard prediction error, and $s(\alpha)$ is the so-called structural complexity which measures the complexity of a program. The design of the DSL and structural cost $s(\alpha)$ is domain-specific, and the goal is to learn an accurate yet simple program. For differentiable programs with a fixed architecture $\alpha$, optimizing the parameters $\theta$ would look like standard neural network training.[1]

**Synthesis Approach**. We search for program architectures using the neurosymbolic synthesis technique known as NEAR [25]. The basic idea is to start with a fully neural program, and search over ways to convert pieces of it into symbolic modules from the DSL. We refer to [25] for more details.

## 4.2 DSL Design

**Design Philosophy.** In program synthesis, a Domain-Specific Language (DSL) is a tailored language designed to generate programs specific to a particular domain. It is important that the construction of our DSL captures features and functions analogous to those employed by domain experts in ECG diagnosis. An expressive DSL ensures enhanced model interpretability; however, the derived program's design should balance depth and simplicity. A shallow structure might inadequately represent the complexity of the problem, whereas an overly complex model may obfuscate interpretability for domain specialists.

**ECG Leads Feature Selection.** A standard 12-lead electrocardiogram (ECG) offers a detailed perspective of the heart's electrical behavior, with each lead distinctly associating to a specified cardiac region as seen in Figure 2. All 12 leads are available as a feature select to the DSL, allowing for the program to focus on specific regions of the heart.

**Engineered Features.** For each of the 12-leads we also include the P, Q, R, S, and T peak locations and heights as feature selects. These features are commonly used by cardiologists to diagnose myocardial infarction and other cardiac abnormalities making them interpretable for clinicians.

---

[1]If $\alpha$ corresponds to the architecture of a standard neural network (e.g., a series of matrix multiplications and non-linearities), then $\theta$ would be all the weights of the network.

Additionally, many other features can be constructed from these five such as QT segment length and RR interval. The full list of DSL features are in Table 5.

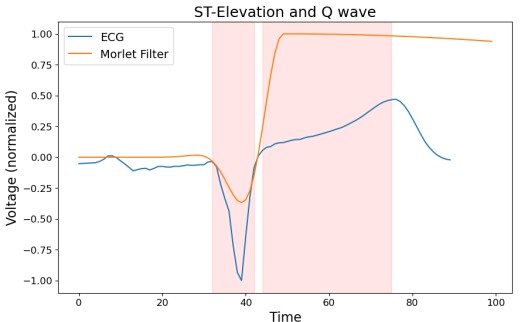 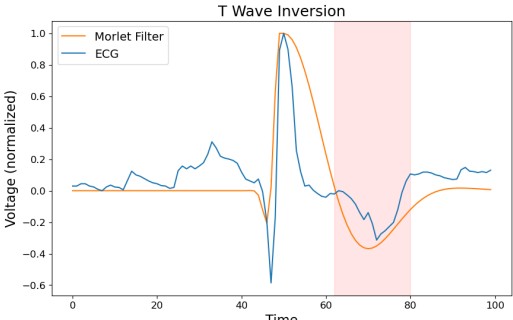

(a) Learned Morlet filter superimposed onto a Lead I heartbeat showing T Wave inversion. The highlighted area shows the location of the morphology.

(b) Learned Morlet filter shown on an example of a Lead V2 heartbeat with Q wave presence. The highlighted area shows the location of the morphology.

Figure 4: Asymmetric Morlet Filter Examples

**Asymmetric Morlet Filter.** To capture specific morphologies within heartbeats, our DSL integrates an asymmetric Morlet filter operation. This operation maps a sequence of vectors to a single vector by computing a weighted sum of the input sequence. The asymmetric Morlet filter, represented by $\psi$, initiates a one-to-one mapping between timesteps $1, \ldots, n$ in the isolated heartbeat to values $x_1, \ldots, x_n$, where $x_i \in [-\pi, \pi] \, \forall i = 1, \ldots, n$. The filter is then evaluated at each $x_i$ and is calculated as:

$$\psi(x; s_1, w_1, s_2, w_2) = \begin{cases} e^{-0.5\left(\frac{x}{s_1/w_1}\right)^2} \cos(w_1 x) & \text{if } x < 0 \\ e^{-0.5\left(\frac{x}{s_2/w_2}\right)^2} \cos(w_2 x) & \text{if } x \geq 0 \end{cases} \tag{1}$$

where $x \in [-\pi, \pi]$. This asymmetric configuration, with parameters $s_1, w_1$ for the left side and $s_2, w_2$ for the right side, gives our DSL the capability to distinguish morphological features endemic to either side of a heartbeat, such as ST elevation and T wave inversion.

We visualize a few examples of learned Morlet filters superimposed on example ECGs (Figures 4a, 4b). Figure 4a shows a learned asymmetric Morlet filter being applied to Lead I of an ECG. The shape of the Morlet filter captures the T wave inversion, and the filter overlaps with the waveform with a high correlation, indicating the presence of T wave inversion in the ECG. Figure 4b shows an example of another learned filter applied to lead V2. The filter selects for ST elevation but also matches closely with the large Q waves present in this lead.

In our framework, the asymmetric Morlet filter is applied to a lead using a cross-correlation operation. Cross-correlation effectively measures the degree of similarity between a signal and a filter at each point of overlap as the filter slides across the signal. The greater the similarity between the signal and the filter, the higher the output of the cross-correlation will be at that point. An example of an asymmetric Morlet filter being used to distinguish ST elevation is shown in Figure 1. The filter applied to lead III is highly activated by patient 2's ECG due to the large degree of overlap between the filter and the ECG waveform. In comparison, lead III in patient 1 shows little similarity to the filter shape and thus has lower levels of activation.

**Basic Functional Components.** In addition to the temporal features and ECG leads described above the DSL also includes basic compositional functions such as "or", "and", smoothed if/then, and map. These functions allow for structurally complex programs using the leads described in Figure 2, the locations of peaks in each ECG channel, and the learned Morlet filters.

## 4.3 Intuition & Recapitulation

To summarize, models or programs that are inherently interpretable must enjoy some form of parsimony. We rely on two forms of parsimony. The first is sparsity, as used in feature selection (choosing a sparse set of leads) and the overall architecture of the program (not every component is densely connected to every other component). The second is domain specific features that enable

capturing rich semantics with only a few variables, such as using Morlet filters rather than more generic convolutional kernels (as used in CNNs). As we will see in the experiments, Morlet filters have (almost) the same representational power as general CNNs for ECG processing, while being directly interpretable in which parts of the ECG waveform are being focused on. Moreover, simple or sparse combinations of Morelet filters remain interpretable. As such, the overall program architecture is sparse enough to be interpreted as a flow-chart (as we visualize in Figures 1, 3, 5).

## 5 Experiments

### 5.1 Evaluation Procedure

For each of the three tasks (Binary, Coarse, Fine) we compare the performance of our learned program to the following two baseline models: XGBoost, a popular gradient boosted decision tree model; and a 1D Convolutional Neural Network.

**XGBoost.** XGBoost is a gradient boosting framework used for traning tree ensemble models. As input to XGBoost we handcrafted several features from the 12 ECG channels, including the P, Q, R, S, T peaks of each beat and specific interval and segment measurements.

**1D Convolutional Network (1D CNN).** A 1D CNN unlike a Morlet Filter, which imposes specific temporal structures on weights, learns unconstrained weights to convolve across input features temporally. The convolution operation aggregates these features and learns to isolate important morphologies from the input features. The output of the convolutional layers will then be fed to the fully connected layers of the network and will contribute to the final output prediction.

**Evaluation.** Models are evaluated using F1-score, calculated as the harmonic mean of Precision and Recall. All models will also be trained on 100% of the training data. The validation and test set are sampled to contain the same distribution of labels as the train set.

### 5.2 Results

**Accuracy.** The programs produced by our approach generally perform better than XGBoost, and is comparable to the black-box 1D CNN (Table 2). The performance gap for the fine-grain (17 locations) classification task is largest, compared to the binary and coarse-grained tasks. We note that our method achieves comparable performance to 1D CNNs while learning a more interpretable structure. Comparing the model performances by location (Figure 6), we observe a similar trend, in that 1D CNNs generally perform the best, followed by our learned programs.

**Interpretability**. We next visualized the structure of several of our learned programs. In Figure 1, we show a binary program trained to identify the presence of IMI, a type of infarction that occurs in the RCA. The program learns to select leads II, III, and aVL. This choice of leads is significant as they are all leads which are well known to be significantly affected by the presence of IMI. Furthermore,

Table 2: F1 score averaged across locations.

|  | **1D CNN** | **Ours** | **XGBoost** |
| --- | --- | --- | --- |
| Binary | 0.84 | 0.83 | 0.76 |
| Coarse | 0.75 | 0.71 | 0.62 |
| Fine | 0.64 | 0.58 | 0.54 |

the asymmetric Morlet filter corresponding to each lead learns to select for a morphological feature. For example, the filter applied to lead III aligns with waveforms showing signs of ST segment elevation. In patient 2 we can see that this filter is highly activated by the ST elevation present in lead III, and the cross correlation score measuring the similarity between the filter and the lead is much higher than the score in the healthy patient (7.5 vs 2.2). The filter applied to lead aVL is most highly activated by the presence of ST segment depression, which is another common sign of IMI when present in lead aVL.

Figure 5 depicts the flow of three ECGs through a program trained on the coarse grain task. The three locations shown are LAD, LCX, and RCA. The leads whose Morlet filters were most highly activated corresponds to the location of the infarction. For example, in the ECG with LCX infarction, the filter applied to Lead I is the most highly activated and has the largest contribution to the final decision of the program. Lead I is a lateral lead (2), and we expect that the filter on this lead should be highly activated when passed an ECG showing signs of LCX infarction; which commonly occurs in the

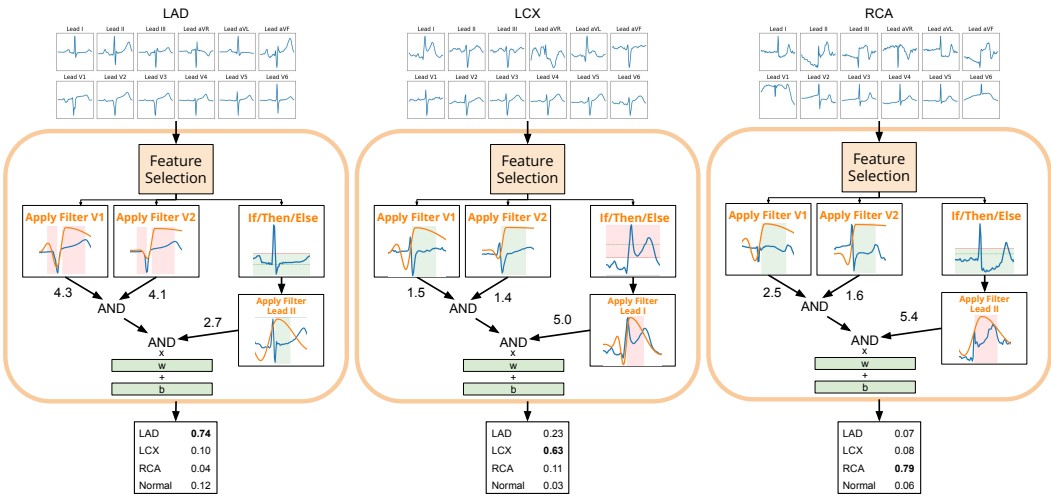

Figure 5: Visualizations of example ECGs being passed through our coarse grain program. Each ECG is taken from one of the respective locations in the coarse grain task: Left Anterior Descending Artery (LAD), Left Circumflex Artery (LCX), Right Coronary Artery (RCA). The program structure and morlet filters are displayed the same as in Figure 1 with green denoting low activation and red high activation.

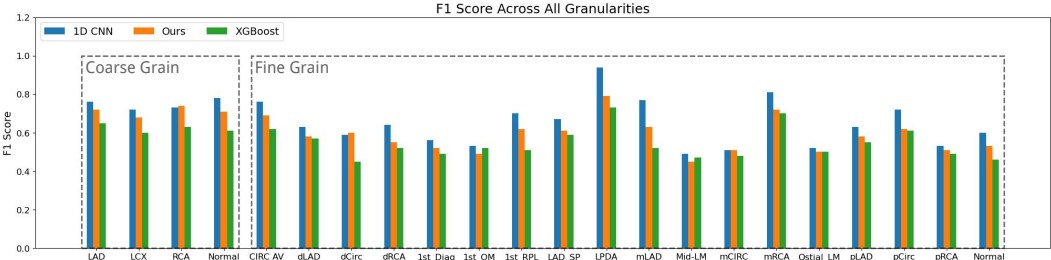

Figure 6: F1 Score by location for coarse and fine grain tasks.

left ventricular lateral wall. The other lead selections correspond to the LAD and RCA locations; besides lead I our program selects leads V1, V2, and II. V1 and V2 correspond to the septal and right ventricle which contain part of the LAD, and Lead II primarily observes the inferior wall of the left ventricle which is where the RCA is located. Figure 4b shows a zoomed in view of the filter applied to lead V2 in our LAD example. In this view, we can see that the filter selects for ST elevation but also matches closely with the large Q waves present in this lead; another sign of MI. We also observed that our learned morlet filter is able to capture signs of T wave inversion (Figure 4a).

*Expert interpretation.* Many of the features human clinicians rely on are recapitulated by the program learning approach. Localization of the myocardial region related to the coronary artery is encoded by the selection of relevant leads optimized during program synthesis. The Morlet filters are able to quantitatively assess relevant patterns such as T wave inversions, Q waves, and ST changes. To a cardiologist, the most relevant features identified by NEAR are similar to the features physicians use to make clinical judgements.

Table 3: DSL Variations F1 Score Averaged Over 10 Runs.

|                  | Binary     | Coarse     | Fine       |
|------------------|------------|------------|------------|
| Original         | 0.83 ±0.01 | 0.71 ±0.01 | 0.58 ±0.02 |
| No Peaks         | 0.76 ±0.01 | 0.64 ±0.02 | 0.49 ±0.03 |
| Symmetric Morlet | 0.81 ±0.01 | 0.67 ±0.01 | 0.55 ±0.03 |
| Running Average  | 0.72 ±0.02 | 0.59 ±0.03 | 0.48 ±0.04 |

**DSL Variations.** To better study the importance of each function in our DSL, we evaluated our approach with several variations (Table 3). Removing handcrafted features (P,Q,R,S,T peaks) from the DSL significantly decreased the performance of the learned programs. This suggests that without

the handcrafted features the synthesized programs are unable to learn peaks location by themselves. Additionally, with the exclusion of handcrafted features the complexity of learned programs was significantly reduced. We also tested our DSL with different temporal filters in place of the asymmetric Morlet filters, such as a symmetric Morlet filter and a running average filter. We found that both of these variations resulted in a decrease in performance. See Appendix D for more details on the DSLs.

# 6 Discussion

Results demonstrate that our synthesized programs perform comparably to neural networks with a much simpler and more interpretable structure. For example, in Figure 1, most of the morlet filters selects for ST elevation, which is the primary marker of MI. We observe that our programs selects leads corresponding to different MI locations in the heart and also learns interpretable morphologies.

*Limitations and next steps.* One limitation of our approach is that modern program synthesis approaches often have a gap in performance to black-box networks, as we observed in our experiments. We note that neurosymbolic program synthesis is an active area of research, and our framework is compatible to new approaches as they develop. Additionally, the performance and interpretability of programs is influenced by the choice of DSL. While we found that our DSL designed with cardiologists was effective, automated methods to scale the design of DSLs across tasks and domains is currently an open question in neurosymbolic modeling.

A future direction that we are excited to explore is to expand our framework to encompass tasks beyond MI localization. The method shown in this paper could be readily adapted to the diagnosis of other types of heart ailments such as ischemia, atrial arrhythmias, and chamber enlargements. Critical to this process is the involvement of domain experts in the design of DSLs. A well crafted DSL can greatly improve the transparency of decision-making; one of the greatest advantages of neurosymbolic methods. As the design of DSLs becomes more accessible to domain experts and as program synthesis technology advances, inherently interpretable frameworks like ours may become invaluable to physicians.

# 7 Conclusion

We propose a framework to learn expert-interpretable programs for localizing myocardial infarctions to specific areas of the heart. Our method is instantiated using a modern program synthesis technique and we demonstrate that our learned programs perform comparably to baseline methods for ECG diagnosis, while being significantly simpler in structure. Our method's capacity to accurately identify the relevant leads and features, which in turn allows for localization of myocardial infarction to specific arteries, demonstrates the potential for assisted clinical diagnostics. Such a system could save clinicians time and offer a valuable second opinion when interpreting healthcare data.

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

# Appendices

## Appendix A    Differentiable Program Synthesis

Our approach uses differentiable program synthesis to automatically learn the program structure $\alpha$ and program parameters $\theta$. We use the recently proposed differentiable program synthesis algorithm NEAR [25], which aims to find programs that minimize the combination of structural cost $s$ and supervised prediction error $\zeta$.

$$(\alpha^*, \theta^*) = \text{argmin}_{(\alpha, \theta)}(s(\alpha) + \zeta(\alpha, \theta)). \tag{2}$$

In this formulation, deeper programs have larger $s(\alpha)$, and the prediction error $\zeta(\alpha, \theta)$ is between the ground truth and predicted classes. The tree in Figure 9 shows an abbreviated view of the program synthesis process. Since the DSL is fully differentiable, NEAR uses neural networks to fill in incomplete programs during the search process. NEAR shows that the performance of these neurosymbolic programs underestimates the total path costs of descendent programs and can be used as an admissible heuristic during search. This allows NEAR to be used with search algorithms such as A* search, which is used in our work.

## Appendix B    Implementation Details

### B.1    Preprocessing

## Preprocessing

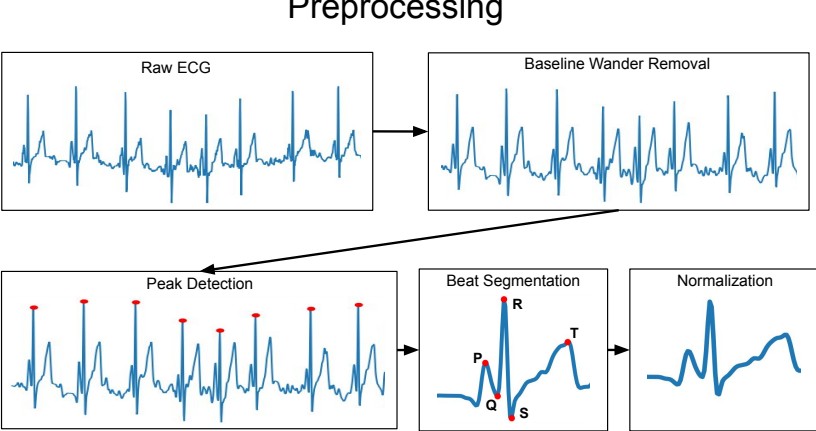

Figure 7: An example of the preprocessing procedure for a single lead. Each lead is filtered to reduce noise and correct baseline wander. Next the P, Q, R, S, and T point of each beat are detected. These points are used to segment the lead into individual beats. A representative beat is then chosen and normalized.

Raw ECGs underwent a series of preprocessing steps (Figure 7). These signals were first denoised with a high-pass Butterworth filter and adjusted for baseline wander. Key waveform features, namely the P, Q, R, S, and T points, were extracted using the QRS detection method described in [19]. Subsequent to these operations, individual heartbeats were isolated from the ECG, normalized, and resized to 100 timesteps. These processed beats were then augmented with the P, Q, R, S, and T points as additional channels in the timeseries data.

### B.2    Hyperparameters

**Program Synthesis.**    The hyperparameters for our program learning approach are in Table 4. Structural penalty and maximum depth are important hyperparameters specific to our program synthesis approach. The structural penalty hyper parameter controls the trade off between program complexity and expressiveness. The maximum depth hyperparameter controls how deep in the search tree NEAR can venture to build programs.

| | n. epochs | s. epochs | frontier size | penalty | max depth | lr | batch size |
|---|---|---|---|---|---|---|---|
| NEAR | 8 | 12 | 20 | 0.001 | 6 | 0.0001 | 64 |

Table 4: Hyperparameters for program learning. n. epochs and s. epochs represent the number of neural and symbolic epochs respectively, where the neural epoch is for the neural heuristic. lr is the learning rate.

**XGBoost.** Our XGBoost implementation used a variety of handcrafted features derived from the 12 ECG channels. These included the P, Q, R, S, and T peaks of each beat as well as the QRS complex dimensions (height and width), QT and PR segment lengths, and RR interval, resulting in a total of 120 features per ECG sample: 12 channels * (5 peaks + 2 QRS dimensions + 2 segments lengths + 1 interval). **1D CNN.** Our implementation consists of three convolutional layers with filter progression of sizes 64, 128, and 256. These layers are followed by a max-pooling layer and subsequently, three fully connected layers. For each of the three tasks the final layer will terminate in either 1 (binary), 18 (fine-grain), or 4 (coarse) neurons depending on the granularity of the task.

## Appendix C Fine Grain Locations

Table 6 shows the list of 17 locations within the heart used for the fine grain task. Most of the locations are rare with the pLAD, mLAD, pRCA, mRCA, and pCirc making up the majority of the ECGs with MI. The locations comprising 1% of the data contain as few as 120 ECGs. Any locations with fewer than this amount were omitted from the fine grain task.

## Appendix D DSL Details

With symmetric morlet filters the programs discovered by NEAR were structurally similar to those discovered with an asymmetric morlet filter. However, the F1 score of the programs for both the coarse and fine grain task dropped. We next tested a much simpler running average temporal filter with window size 13. The program structures produced by NEAR were much simpler than when using either morlet filter as seen in Figure 8; suggesting that a running average is not complex enough to capture important features from the ECG waveforms. Additionally, the performance of the resultant programs suffered on all three tasks.

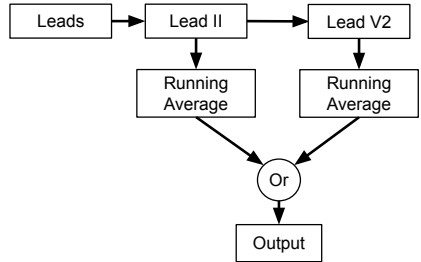

Figure 8: Program produced on coarse classification task by NEAR using running average filter.

|                | Full DSL | No Peaks DSL |
|----------------|----------|--------------|
| List-to-List   | Map, If-Then-Else | Map, If-Then-Else |
| List-To-Atom   | Or, And, Asymmetric Morlet Filter | Or, And, Asymmetric Morlet Filter |
| Atom-to-Atom   | Lead Selects: I, II, III, aVF, aVR, aVL,V1, V2, V3, V4, V5, V6 Peak Selects: P, Q, R, S, T | Lead Selects: I, II, III, aVF, aVR, aVL,V1, V2, V3, V4, V5, V6 |
|                | Symmetric Morlet DSL | Running Average DSL |
| List-to-List   | Map, If-Then-Else | Map, If-Then-Else |
| List-To-Atom   | Or, And, Morlet Filter | Or, And, Running Average 13 |
| Atom-to-Atom   | Lead Selects: I, II, III, aVF, aVR, aVL,V1, V2, V3, V4, V5, V6 Peak Selects: P, Q, R, S, T | Lead Selects: I, II, III, aVF, aVR, aVL,V1, V2, V3, V4, V5, V6 Peak Selects: P, Q, R, S, T |

Table 5: Full DSLs for each variation in Table 3.

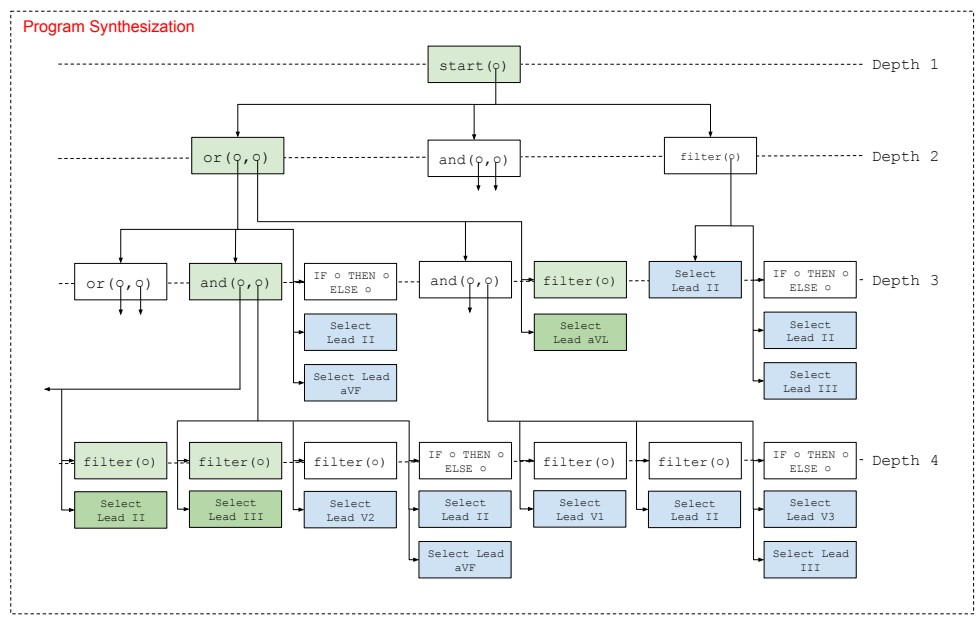

Figure 9: Differentiable program synthesis search tree. Green nodes represent the path chosen by the search algorithm for the final coarse grain program.

| Location | Label Distribution (%) |
|---|---|
| Circumflex AV groove (CIRC AV) | 2 |
| Distal Left Anterior Descending (dLAD) | 3 |
| Distal Circumflex (dCirc) | 3 |
| Distal Right Coronary Artery (dRCA) | 4 |
| First Diagonal (1st_Diag) | 2 |
| First Obtuse Marginal (1st_OM) | 2 |
| First Right Posterolateral (1st_RPL) | 1 |
| Left Anterior Descending Septal Perforator (LAD_SP) | 1 |
| Left Posterior Descending Artery (LPDA) | 1 |
| Mid Left Anterior Descending (mLAD) | 6 |
| Mid Left Main (Mid-LM) | 2 |
| Mid Circumflex (mCIRC) | 4 |
| Mid Right Coronary Artery (mRCA) | 5 |
| Ostial Left Main (Ostial_LM) | 1 |
| Proximal Left Anterior Descending (pLAD) | 9 |
| Proximal Circumflex (pCirc) | 5 |
| Proximal Right Coronary Artery (pRCA) | 7 |
| Normal | 42 |

Table 6: Fine Grain Task Location and Label Distributions.

