# OpenReview forum: "Learning Expert-Interpretable Programs for Myocardial Infarction Localization"
_NeurIPS.cc/2023/Workshop/AI4Science — NeurIPS2023-AI4Science Poster_

### Official Review · Reviewer_csea · 2023-10-25
**MI Localization**

**Rating:** 7
**Confidence:** 5

**Review:**

The authors present a detailed algorithim for detecting myocardial infarctions (MI) from electrocardiogram (ECG) signals. The use of asymmetric Morelet filters for crosscorrelation analysis was very practial. The breakdown of sub-groups to binary, coarse, and fine shows the level of granularity introduced by the authors in the study.

- Since there were more than 20k samples, why was leave-one-out cross-validation not considered during classification?
- Furthermore, the patient demographic includes a lot of cardiovascular pathologies. What would be the expected outcome if the data set comprises of patients with no cariovascular diseases?
- It was suprising to see that XGBoost underperforming in this scenario. Since the data is complex, would AdaBoost work better?

---

### Meta-Review · Area_Chair_V2U9 · 2023-10-27

**Recommendation:** Accept (Poster)
**Confidence:** 4

**Metareview:**

Authors have proposed an expert-interpretable program for myocardial infarction localization from ECG signals by applying recent algorithms from neurosymbolic learning. These algorithms use a domain specific language which authors developed with help of experts. Overall, the paper is written clearly, and presents a novel perspective to the field as there is huge research on using ECG which is mostly focused on deep learning technqiues. However, empirical evaluation is weak as they used very basic baselines and did not compare against the SOTA.
First, interpretability is important for the adoption of AI models and second this paper also employs a novel perspective so I recommend acceptance.